# Adaptive variation in avian eggshell gas conductance and structure across elevational gradients?

David Ocampo[1,2]*, Carlos Daniel Cadena[2], Esteban Correa-Agudelo[3,4], Marcela Hernández Hoyos[3], Gustavo A Londoño[5]*

[1]Department of Ecology and Evolutionary Biology, Princeton University, Princeton, United States; [2]Laboratorio de Biología Evolutiva de Vertebrados, Departamento de Ciencias Biológicas, Universidad de Los Andes, Bogotá, Colombia; [3]Departamento de Ingeniería de Sistemas y Computación, Universidad de los Andes, Bogotá, Colombia; [4]Division of Asthma Research, Department of Pediatrics, Cincinnati Children's Hospital Medical Center, University of Cincinnati College of Medicine, Cincinnati, United States; [5]Departamento de Ciencias Biológicas, Bioprocesos y Biotecnología, Universidad Icesi, Cali, Colombia

*For correspondence:
docampo@princeton.edu (DO);
galondono@icesi.edu.co (GAL)

**Competing interest:** The authors declare that no competing interests exist.

**Abstract** Many tropical bird species have restricted elevational distributions, potentially limited by how environmental conditions affect physiological processes. While some studies have examined adult physiology across elevations, relatively little attention has been given to the structure and function of eggshells despite their critical role in regulating gas exchange during the vulnerable embryonic stage. At high elevations, dry air is expected to increase water loss from the egg, and natural selection may favor lower gas conductance to reduce desiccation risk. Structural variation in eggshells, such as increased shell thickness or reduced pore size and density, could serve as a mechanism to regulate gas diffusion. To test for adaptive variation in eggshell traits along elevational gradients, we measured water vapor conductance and used scanning electron microscopy (SEM) to examine eggshell structure in 197 bird species from the Andes. We found that water vapor conductance declined at high elevations across avian communities. However, structural changes in eggshells varied among bird families and did not vary in a predictable way with elevation, suggesting no relationship or divergent adaptive responses to shared selective pressures, particularly in shell thickness, pore density, and pore size. We propose that examining functional and structural eggshell traits can offer insight into species' elevational limits and inform predictions about their responses to climate change.

## Editor's evaluation

This important study provides solid evidence that high-elevation species lose water at slower rates than low-elevation species. The findings imply that egg physiology may be a factor limiting the distributions of bird species. This work reinforces the need for all life stages to be considered when evaluating physiological adjustment to climate change.

## Introduction

Species distributions along environmental gradients are determined by both biotic interactions and abiotic conditions (e.g. temperature, humidity). The role these factors play in shaping the elevational distributions of tropical montane species has received increasing attention, partly due to a growing

**eLife digest** Along mountain slopes, changes in humidity, air pressure, and temperature can affect physiological processes and constrain where species can live. As a result, plants and animals are often adapted to the specific conditions of their environments. These patterns have been widely studied across organisms, including birds, where researchers have examined metabolic rates and insulating features such as adult plumage. However, less is known about an even more vulnerable stage of life – when individuals cannot move, and development has only just begun.

During embryonic development within an egg, a bird depends on a delicate balance with the external environment. Even small environmental changes can disrupt this balance by altering gas exchange through microscopic pores in the eggshell. Oxygen must diffuse inward, while carbon dioxide and water vapor diffuse outward. At high elevations, maintaining this balance becomes more challenging because dry air increases the risk of excessive water loss. Studying eggshell adaptations can provide refined insights into how environmental conditions during the nesting period may constrain species distributions.

Ocampo et al. asked whether birds in tropical mountains cope with environmental challenges during nesting by adjusting gas exchange through the eggshell, particularly by reducing water loss at high elevations. They then investigated the mechanisms underlying these adjustments by examining variation in eggshell microstructure across many species, including traits such as thickness and porosity.

First, the researchers documented several nests and eggs not previously described by science, particularly from species inhabiting remote regions of the Amazon and the Andes. By examining egg physiology along elevational gradients, they found that high-elevation species lose water at slower rates than lowland species, likely as an adaptation to reduce the risk of desiccation. These findings support the idea that egg physiology may constrain species distributions. They also identified microstructural variation in eggshells across lineages, although further work is needed to clarify the patterns and their functional significance.

Overall, the study of Ocampo et al. highlights the importance of considering all life stages when evaluating physiological responses to climate change. Studies of species' natural history remain essential and, when combined with hypothesis-driven evolutionary ecology, provide a powerful framework for understanding species distributions and improving predictions of their responses to environmental change. These findings may inform ecologists and physiologists, as well as conservation practitioners, land managers, policymakers, and researchers developing bioinspired materials. Realizing these benefits will require translating findings into predictive models and management strategies, alongside further advancing our understanding of eggshell structure.

interest in forecasting species' responses to climate change (*Sheldon et al., 2011*; *García-Robledo et al., 2016*; *Zuloaga and Kerr, 2017*; *Freeman et al., 2018*; *Bota-Sierra et al., 2022*). Tropical birds, which often exhibit narrow elevational ranges and abrupt replacement by closely related species along mountain slopes, are a model system for studying elevational range limits (*Terborgh, 1971*; *Sheldon et al., 2011*; *Freeman et al., 2022*). Beyond biotic interactions, physiological constraints in adults, such as those related to heat, energy balance, and respiration, likely also set elevational limits in tropical birds (*Janzen, 1967*; *Ghalambor et al., 2006*; *Cadena and Loiselle, 2007*; *Storz et al., 2010*; *Londoño et al., 2015*; *Londoño et al., 2017*). A complementary hypothesis is that the physiological requirements of embryos may also limit birds' ability to breed across broad elevational ranges (*Jankowski et al., 2013a*) because the partial pressure of oxygen, humidity, and temperature, which vary with elevation, are key variables influencing embryo development (*Carey et al., 1982*; *Deeming, 2002*; *Deeming, 2011*).

One major factor mediating embryonic responses to environmental conditions is eggshell structure, which plays a central role in regulating gas exchange and water retention during development. During development, the exchange of oxygen and water vapor in bird eggs occurs via diffusion across the shell through microscopic pores (*Tullett and Board, 1977*). Consequently, the evolution of avian eggshell structure may reflect a trade-off between oxygen uptake and water loss, including the risk of embryonic desiccation, both of which are influenced by eggshell thickness and the density and size of

pores (*Ar et al., 1974*; *Vleck et al., 1979*). While eggshell structure has been studied in a few model species (*Rahn et al., 1979*), particularly in relation to egg size, little is known about how eggshell traits vary across bird species along environmental gradients (but see *Sotherland et al., 1980*; *Board, 1982*; *Carey et al., 1989*; *Stein and Badyaev, 2011*). Similarly, although there is substantial research on the factors influencing gas exchange across eggshells (*Sotherland et al., 1980*; *Vleck et al., 1983*), comparative analyses aimed at understanding how environmental conditions drive interspecific variation in eggshell function increased in the last decade (*Portugal et al., 2014*; *Attard et al., 2021a*; *Attard and Portugal, 2021b*; *Attard and Portugal, 2022*).

Studies on temperate-zone species have shown that avian eggshell structure varies with environmental conditions such as temperature, humidity, and barometric pressure. In particular, bird species inhabiting dry, cold, high-elevation environments tend to have eggs with lower pore density or pore size and increased eggshell thickness, traits that reduce gas diffusion compared to those of lowland species (*Wangensteen et al., 1974*; *Rahn et al., 1979*; *Rahn and Ar, 1980*; *Sotherland et al., 1980*; *Carey, 1980a*; *Carey et al., 1983b*; *Carey et al., 1984*; *Rahn et al., 1982*; *Rahn and Paganelli, 1990*). In contrast, a population of a passerine species that recently colonized a high-humidity environment produced larger eggs with thicker shells and lower pore density than populations from drier habitats (*Stein and Badyaev, 2011*). This pattern runs counter to expectations if gas exchange were the primary selective pressure shaping eggshell traits. Instead, the observed variation may reflect an elevated risk of trans-shell bacterial infection in humid environments (*Cook et al., 2005*).

What little is known about eggshell structure in tropical birds has largely been generalized from studies of a limited number of taxa. For example, among ducks and grebes, high-elevation species tend to have eggshells with lower pore density and reduced water vapor conductance compared to closely related lowland species of similar size (*Carey et al., 1982*; *Carey, 1983a*; *Carey et al., 1990*). In this study, we apply phylogenetic comparative methods to water vapor conductance and scanning electron microscopy (SEM) data to examine patterns of variation in eggshell function and structure across elevational gradients in a diverse set of Neotropical landbirds. Atmospheric pressure and water vapor pressure both decline with elevation, such that, all else being equal, eggs are expected to lose water more rapidly at higher altitudes (*Romanoff, 1930*; *Wangensteen et al., 1974*). We therefore tested the prediction that eggshells of highland birds should exhibit reduced water vapor diffusion rates to retain moisture throughout the incubation period (*Sotherland et al., 1980*; *Rahn et al., 1982*). This could be achieved by producing eggs with thicker shells (i.e. longer pores), smaller pore diameters, or lower pore densities compared to those of lower elevation species (*Wangensteen et al., 1970*; *Board, 1982*). We also asked whether eggshell function and structure vary adaptively within species due to mechanisms similar to those acting across species. By focusing on a diverse and unexplored set of Neotropical landbirds, we aim to evaluate variation in eggshell traits that may contribute to species' reproductive success across elevational gradients.

## Results
### Conductance

The average eggshell water vapor permeability, measured as conductance ($G_{H2O}$) for all bird species included in our analyses was $1.2\pm1.7$ mg·day$^{-1}$ torr$^{-1}$, ranging from sparrows (*Zonotrichia capensis*) with 0.1 mg·day$^{-1}$ torr$^{-1}$ to tinamous (*Tinamus major*) with 12.3 mg·day$^{-1}$ torr$^{-1}$ (*Figure 1a*). We found low phylogenetic signal in eggshell conductance ($G_{H2O}$; Pagel's $\lambda$ =0.38), indicating that this trait is evolutionarily labile. The OU model provided the best fit to our data and was used for all subsequent comparative analyses (*Supplementary file 4*). The best-supported model ($\Delta AIC \leq 2$) included elevation and egg weight (EW) as predictors ($G_{H2O}$ ~ 1 + Elevation+EW; AICc = 834.25, model weight = 0.6461). Larger eggs exhibited greater water loss than smaller eggs, while eggs from highlands lost less water compared to those from mid-elevation and lowlands (slope = 0.56, $R^2$=0.27, p<0.001; *Figure 1b*), consistent with predictions regarding elevational effects on gas exchange. Since large eggs (i.e. >12 g) are only present at mid-elevations and in the lowlands, we conducted the same analysis and found consistent patterns (see *Figure 1—figure supplement 1*). This indicates that the overall pattern was not driven by elevational differences in egg size.

After correcting for egg mass, residual $G_{H2O}$ differed significantly among stations (phylogenetic ANOVA; $F_{2,108}$ = 29.77, p<0.001), decreasing with increasing elevation. Additionally, the variance in

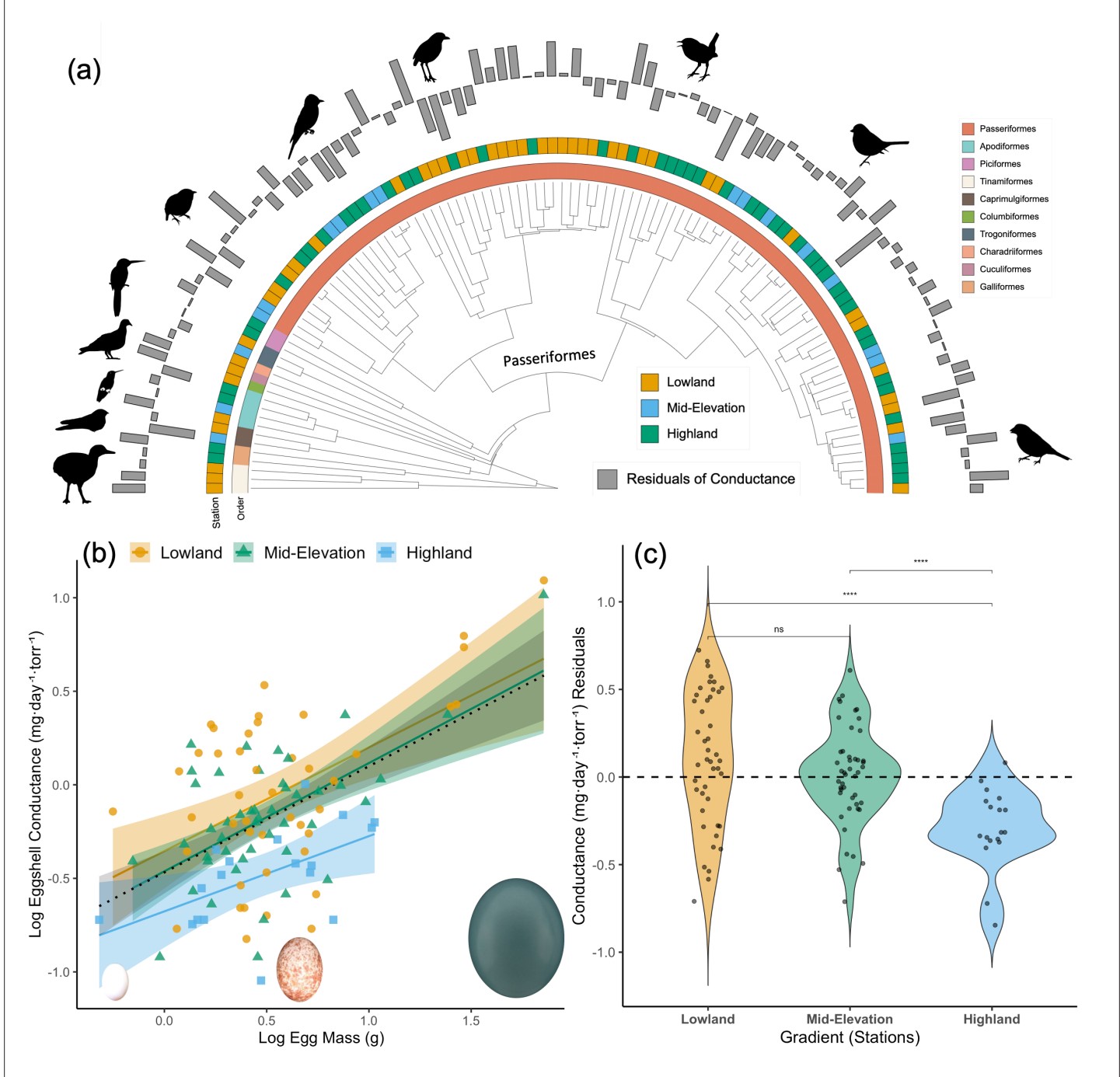

**Figure 1.** Variation in eggshell conductance across Andean bird species. (**a**) Phylogeny of the studied species visualized using iTOL v. 6.7.4. (***Letunic and Bork, 2021***). The internal half-circle represents orders in colors, the external half-circle represents elevations, and the grey bars represent the residual values of conductance. (**b**) The significant positive relationship (*log Conductance = $\beta_0 + \beta_1 \cdot log$ Mass; $R^2 = 0.63$, $p < 0.001$*) between log-transformed egg mass and log-transformed eggshell conductance across 108 species of birds in three elevational zones (Lowland, Mid-Elevation, Highland). Shaded areas represent 95% confidence intervals around the regression lines. Colors and shapes indicate elevation categories; egg photos by DO for *Threnetes leucurus*, *Atlapetes melanolaemus,* and *Tinamus major*. (**c**) Residual conductance of species occupying three different elevational ranges (Phylogenetic ANOVA; $F_{2, 106} = 4.46$, $P < 0.001$; lowland $n = 44$; medium elevation $n = 43$; and highland $n = 18$); significant differences indicated with *. Note that lowland species show wide conductance values with a mean above the expected, given the allometric relationship. Highland species exhibit less variation, and the conductance is lower than expected for the size of their eggs. Bird silhouettes in (**a**) are from www.phylopic.org (*Crypturellus variegatus* by JN Wiegers; *Haplophaedia aureliae* by Edwin Price; *Manacus manacus* by Edwin Price).

*Figure 1 continued on next page*

*Figure 1 continued*

The online version of this article includes the following figure supplement(s) for figure 1:

**Figure supplement 1.** Analysis excluding large eggs (i.e. >12 g), only present at mid-elevations and in the lowlands.

$G_{H2O}$ differed among stations ($F_{2,108}$ = 4.46, p<0.001), with greater variation observed in the lowlands and mid-elevations compared to the highlands (*Figure 1c*).

## Eggshell structure

Shell thickness exhibited a higher phylogenetic signal (Pagel's $\lambda$ =0.48) compared to pore density ($\lambda$ =0.34) and pore size ($\lambda$ =0.18), although values of $\lambda$ for all traits were low (*Figure 2*). This suggests that these eggshell traits are evolutionarily labile (*Figure 3c, f, and i*), potentially due to varying selective pressures across different nesting habitats along the elevational gradient. Repeatability was very high for eggshell thickness (R=0.997, 95% CI=0.996–0.998, LRT p=1.62 × $10^{-286}$), indicating that replicate thickness measurements were highly consistent and that eggshell thickness represents a robust species-level trait. Pore number also showed high repeatability (R=0.797, 95% CI=0.721–0.849, LRT p=2.51 × $10^{-46}$), supporting its use as a reliable species-level variable. In contrast, pore area exhibited no detectable repeatability (R=0, 95% CI=0–0.119, LRT p=1), suggesting that variation in pore area is dominated by within-species heterogeneity or higher measurement variability. Accordingly, we used species-level averages for eggshell thickness and pore number in subsequent analyses, whereas results involving pore area should be interpreted with caution.

Eggshell thickness was positively correlated with egg volume ($R^2$=0.84, p<0.001; *Figure 3a*). Phylogenetic generalized least squares (PGLS) analyses revealed that, across all species, eggshell thickness was not significantly associated with elevation after correcting for egg volume ($R^2$=0.007, p=0.053; *Figure 3b*). Pore density was negatively related to egg volume ($R^2$=0.285, p<0.001; *Figure 3d*). Pore density corrected by egg size did not change significantly with elevation across all species ($R^2$=0.01, p=0.99; *Figure 3e*). Pore size was not significantly correlated with egg volume ($R^2$=0.002, p=0.38; *Figure 3g*) and showed no significant relationship with elevation across all species ($R^2$=0.004, p=0.51). A close exploration among clades with species along the whole gradient revealed a significant positive relationship between eggshell thickness residuals and elevation in the Thraupidae family ($R^2$=0.33, p=0.018, n=14; *Figure 3b*), a negative relationship between residual pore density and elevation in Nightjars (Caprimulgidae; $R^2$=0.87, p=0.014, n=5), and a strong positive relationship between pore size and elevation for the Furnariidae family ($R^2$=0.72, p=0.001, n=10; *Figure 3h*).

Variance within species did not reveal statistically significant differences consistent with adaptation or plasticity in response to elevation. In the *Atlapetes melanolaemus* samples spanning 1000 m in elevation (range = 2300–3000 m), we found no significant relationships between elevation and thickness of the shell ($R^2$=0.004, p=0.33), pore density ($R^2$=0.023, p=0.41), or pore size ($R^2$=0.074, p=0.68). Likewise, there were no significant relationships between elevation and shell thickness ($R^2$=0.04, p=0.20, n=19), pore density ($R^2$=0.07, p=0.6), or pore size ($R^2$=0.06, p=0.6) for the *Pyrrhomyias cinnamomeus* spanning over a 1700 m in elevation (range = 1300–3000 m).

## Discussion

Our analyses of a large sample of Neotropical bird species in Andean areas revealed that water vapor conductance across the eggshell decreases at high elevation, consistent with adaptations for better humidity retention in the dry, cold conditions. But the underlying mechanisms related to eggshell characteristics varied among taxa, resulting in no overall relationship between elevation and eggshell traits. Our within-species analyses revealed some variation at the species level in egg structure in species that occupy a wide elevation range. However, within-species variation was unrelated to elevation. This suggests that variation in eggshell structure within populations may allow species to occur over a broad range of elevations, but it is unclear whether the lack of variation with respect to elevation (hence presumably lack of adaptation at range margins) may contribute to setting elevational range limits.

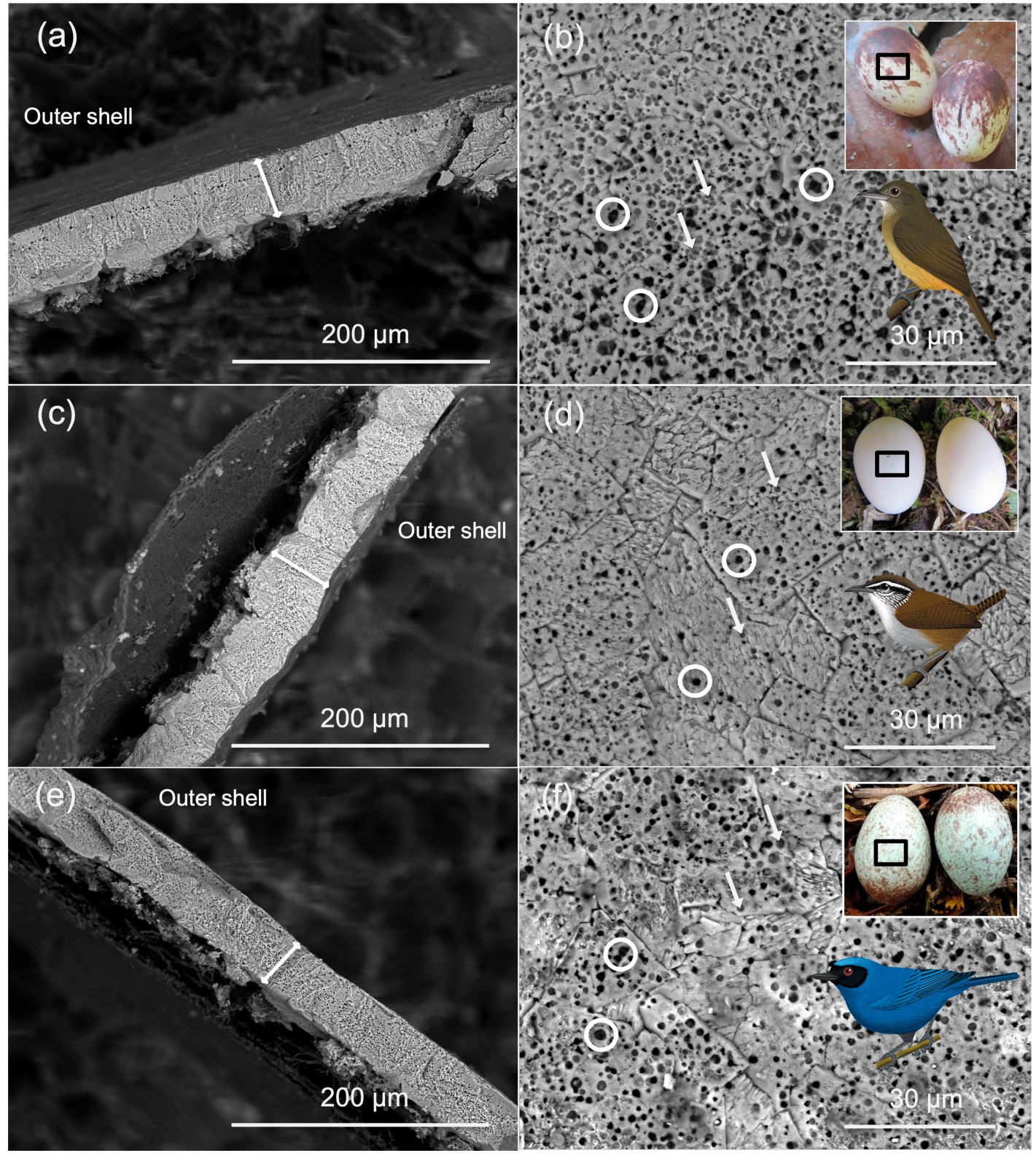

**Figure 2.** Scanning electron microscopy (SEM) images of the cross-section (**a, c, e**) and the eggshell surface and eggs species photo/illustration (**b, d, f**) of Dusky-throated Antshrike (*Thamnomanes ardesiacus,* **a and b**), Gray-breasted Wood-Wren (*Henicorhina leucophrys,* **c and d**), and Masked Flowerpiercer (*Diglossa cyanea,* **e** and **f**). White double arrows in a, c, and e show the thickness measurement taken from the eggshell; white circles in b, d, and f point to exposed pores counted, and white arrows in occluded holes. Images by DO, photos by GAL, and bird illustrations by Fernando Ayerbe-Quiñones.

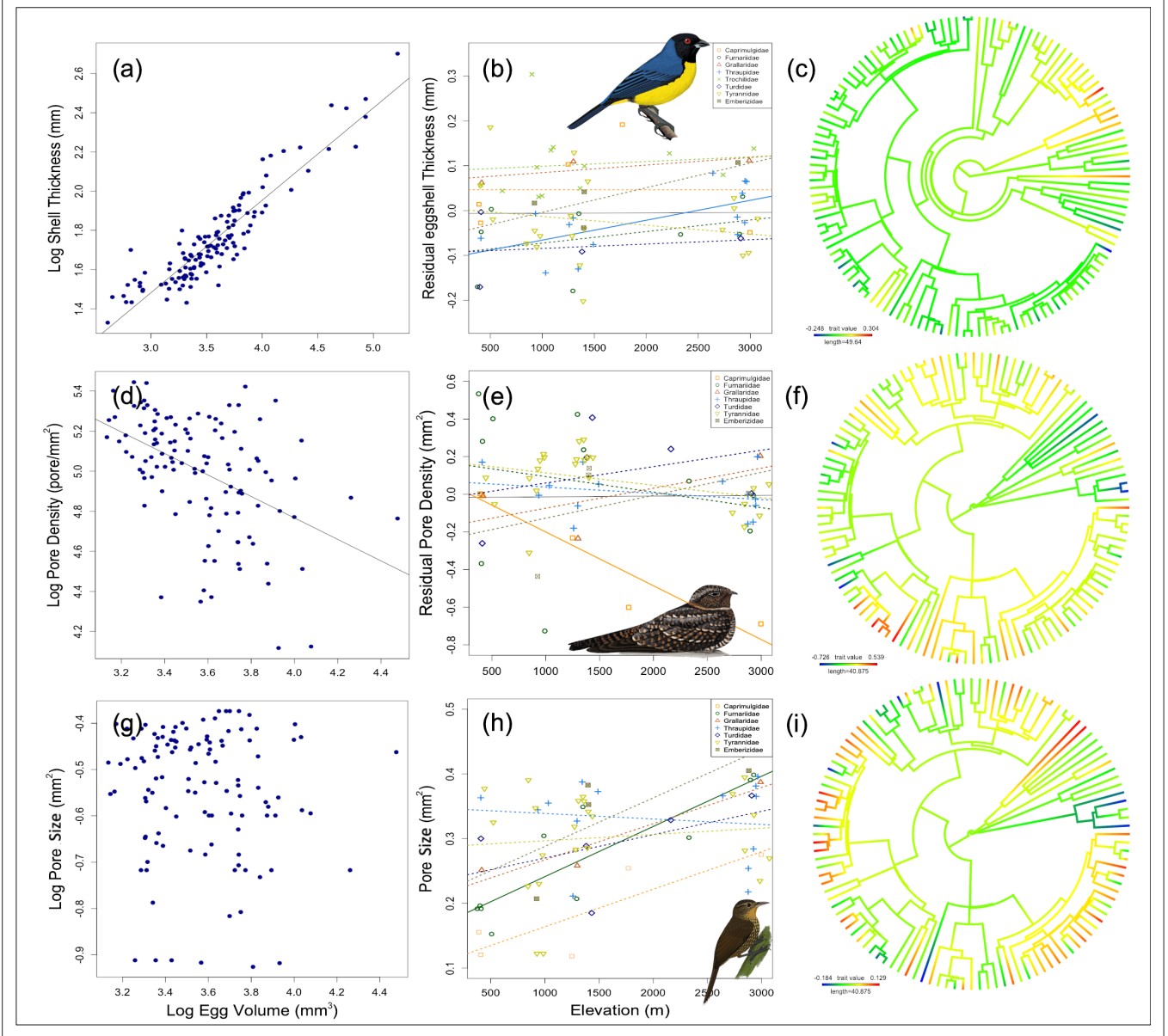

**Figure 3.** Left: Allometric relationship between (**a**) thickness, (**d**) pore density, and (**g**) pore size and egg volume. Note that thickness is positively correlated with egg size (p<0.001, n=129), while pore density and egg size are negatively related (p<0.001, n=116); there is no relationship between pore size and egg volume; therefore, we used absolute values. Center: Relationships between residual (i.e. size-independent) thickness (**b**), pore density (**e**), absolute pore size (**h**), and elevation for families distributed along the gradient. In all cases, there are no overall relationships, but both positive and negative relationships are found when data are analyzed separately for species in different families: statistically significant relationships are shown as solid lines, otherwise as dotted lines. Right: Trait evolution of residual thickness (**c**), pore density (**f**), and absolute pore size (**i**). Bird illustrations by Fernando Ayerbe-Quiñones.

Our main results align with findings from a global study showing lower conductances in species breeding at high latitudes, where preventing dehydration is critical due to temperature fluctuations that can suspend embryo growth and prolong incubation (*Attard and Portugal, 2021b*). Additionally, our data showed significantly larger variation in conductance at lower and mid-elevations, while high elevations exhibited reduced variance, likely reflecting environmental filtering, where only species that can reduce their conductance and maintain it at suitable levels can successfully breed at these elevations (*Rahn et al., 1977*; *Sotherland et al., 1980*; *Carey, 1980a*; *Carey, 1983a*; *Carey et al., 1989*; *Rahn et al., 1982*). Together, these studies underscore the role of environmental pressures in shaping eggshell function across diverse bird lineages.

Alternatively, the higher variation in conductance observed in lowland and mid-elevation species may reflect relaxed selection pressures and the greater diversity of microclimates (associated with greater habitat complexity) at these elevations (*Terborgh, 1977*; *Patterson et al., 1998*). Avian diversity in our study region is also highest at lower elevations (*Terborgh, 1977*; *Jankowski et al., 2013b*), and lowland species exhibit a broad spectrum of life-history traits (e.g. nest construction, nest location, incubation behavior, embryo development period) that could contribute to the wide variation in water vapor conductance (*Deeming and Ferguson., 1991*; *Deeming, 2002*; *Mortola, 2009*; *Attard and Portugal, 2021b*). We acknowledge that sampling was uneven across elevations, reflecting natural patterns of species richness rather than intentional bias. This uneven representation may have limited our ability to detect clear elevational trends in some eggshell traits. However, we retained all available species to maximize taxonomic coverage, particularly because nests and eggs remain poorly known for many lineages. By incorporating phylogenetic corrections in all analyses, we aimed to minimize potential biases arising from non-independence among species.

Birds may achieve reduced conductance at higher elevations through different combinations of thicker eggshells, lower pore density, or reduced pore size. When we explored the pattern focusing on clades with species along the gradient, we identified two primary patterns in eggshell characteristics that could support our hypothesis. In tanagers (Thraupidae), shell thickness increased with elevation, while pore density and area remained constant. In contrast, in nightjars (Caprimulgidae), shell thickness slightly decreased with elevation, while pore density declined. In the opposite direction, in furnariids and spinetails (Furnariidae), pore size tended to increase. These three families differ greatly in basic breeding ecology, particularly in nest type and location (i.e. nesting in cups on vegetated areas, on the ground without a proper nest structure, and in elaborate domes, respectively), which may impose different selective pressures as a result of diverse egg microclimate (*Rahn et al., 1982*; *Portugal et al., 2014*), regardless of elevation. While we did not find an effect of the nest type on conductance, nest characteristics such as materials and location (*Collias and Collias, 1984*; *Hansell, 2000*; *Heenan et al., 2015*; *Gómez et al., 2019*; *Batisteli et al., 2021*; *Attard and Portugal, 2021b*) should be considered to better understand how nest architecture influences the microclimate inside the nest. It is also possible that other egg traits, such as size, influence gas exchange across the eggshell and could vary adaptively with elevation. For example, egg size may be larger in species nesting in cooler environments to reduce heat loss (*Martin, 2008*; *Heming and Marini, 2015*), or egg mass may increase at higher elevations, with a larger investment of water in eggs to compensate for greater water loss during incubation (*Carey, 1983a*). To directly test for within-species adaptation in egg traits to environmental variables varying with elevation, common-garden experiments would be necessary.

In addition to pore structure and shell thickness, several other factors may play a role in balancing gas diffusion and water flux through the eggshell, and these factors may vary among taxa (*Romanoff and Romanoff, 1949*; *Becking, 1975*). Such factors include behavior, attendance during the incubation period (e.g. length and rhythms), and its impact on nest temperatures (*Carey, 1980b*; *Carey, 1983a*; *Lourens et al., 2005*; *Boyle et al., 2016*). Another possibility is that eggshell structure may not always respond to elevation as predicted because humidity inside the nest, the primary driver of variation in eggshell structure (*Walsberg, 1983*; *Carey, 1983a*; *Portugal et al., 2014*; *D'Alba et al., 2016*), does not necessarily vary linearly with elevation. For example, species might regulate nest humidity through architectural features, location, and materials or behavioral adjustments (*Deeming, 2011*). However, detailed information on microscale climatic variation (i.e. inside the nest) is lacking at our study sites. Future research should focus on the microclimate of individual nests to more accurately characterize the environmental pressures related to temperature and water vapor pressure that eggs experience.

The lack of consistent patterns across taxa that we observed may be due to differing selective pressures associated with abiotic factors, driving the evolution of eggshell characteristics in opposite directions. For example, in the temperate zone, low-elevation populations of House Finches (*Haemorhous mexicanus*) exhibit larger eggs with thicker shells and lower pore density compared to populations from higher elevations, resulting in lower conductance in lowland populations (*Stein and Badyaev, 2011*). Based on our hypothesis, we would have expected these eggshell traits to be found in highland species, not in lowland ones. However, in House Finches that inhabit small elevational

gradients, a more significant selective pressure may be the increased risk of trans-shell bacterial infection, which is heightened under higher humidity conditions (*Cook et al., 2005*).

## Methodological challenges

One caveat is that we were unable to maintain a constant temperature across elevations in our field experiments; the reduced conductance in eggs from higher elevations may partly reflect the lower temperatures at these elevations. While the exact magnitude of this effect is unknown, definitive conclusions regarding variation in eggshell conductance with elevation will require additional experiments with closely controlled temperature conditions (see *Carey, 1983a*; *Portugal et al., 2010*). Nonetheless, the observed pattern of greater variation in conductance among lowland and mid-elevation species cannot be solely attributed to temperature differences across our experiments.

Another potential limitation of our study is the incomplete understanding of the functional importance of eggshell structures as quantified via SEM images. For example, we cannot be certain that all the pores detected on the surface penetrate the entire shell. Additionally, our current knowledge of pore structure mainly comes from model species, such as chickens and birds from other non-passerine families (*Board et al., 1977*; *Rahn et al., 1979*; *Tullett, 1984*; *Vieco-Galvez et al., 2021*). We assumed that dark pores penetrate the shell, but we acknowledge that this is not always the case, as substances covering the surface may occlude the pores (*Board et al., 1977*; *Martín-Vivaldi et al., 2014*). Employing techniques that can identify functional pores (e.g. *Hargitai et al., 2011*; *Jaeckle et al., 2012*; *Bowers et al., 2015*) would provide valuable additional insights beyond microscopy. Furthermore, we did not examine the three-dimensional (3D) structure of egg pores, which could influence gas exchange and water loss. However, studies on pore morphology suggest that all species included in our eggshell structure analysis have simple, unbranched canals (*Board et al., 1977*; *Tullett, 1984*; *Murphy et al., 2015*). Future analyses focusing on the 3D cross-structure of the eggshell in small species may help clarify the trade-offs between structural constraints of the eggshell and pore matrix configuration.

## Avian eggshells and elevational distributions

Most studies examining variation in avian life-history strategies along elevational gradients have focused on how abiotic and biotic factors influence variables such as fecundity, incubation behavior, clutch size, embryonic and nestling development, and nesting success across multiple species (*Badyaev, 1997*; *Badyaev and Ghalambor, 2001*; *Boyce et al., 2015*). However, large-scale, cross-species analyses may obscure patterns within clades that have adapted to similar evolutionary challenges in distinct ways (*Boyle et al., 2016*). Understanding the environmental tolerances of species is crucial for unraveling the mechanisms behind species distributions along elevational gradients (*Jankowski et al., 2013a*; *Graham et al., 2014*), though studies on this topic in birds are still limited (*Londoño et al., 2015*; *Londoño et al., 2017*; *Freeman, 2016*). Our work suggests that exploring alternative functional mechanisms, such as reproductive traits, may offer new perspectives on the factors shaping avian distributions across mountains. Our findings support the hypothesis that egg characteristics may have evolved or adjusted through plasticity to align gas exchange rates with local environmental conditions, with different species responding to common pressures in diverse ways (*Board et al., 1977*; *Board, 1980*; *Lourens et al., 2005*). Thus, differences among families in the variation of eggshell structure along elevational gradients may reflect alternative adaptations to reduce desiccation (*Rahn et al., 1977*; *Birchard and Deeming, 2009*). However, the presumed adaptive variation was not universal.

Finally, we propose that further study is needed to assess the role of putative adaptations or the lack thereof in eggshell function and structure in shaping the limits of elevational ranges. Specifically, the need to provide sufficient oxygen for embryonic development while preventing desiccation may constrain bird distributions at higher elevations and influence species' responses to climatic change (*Sheldon et al., 2011*; *Freeman et al., 2018*). Given the narrow elevational ranges of tropical species and their presumably limited physiological tolerances (*Janzen, 1967*), montane species may be particularly vulnerable to shifts in environmental conditions (*Pearson and Dawson, 2003*; *Colwell et al., 2008*). Even slight environmental changes could impact the microclimates where birds breed and harm embryo development (*Lourens et al., 2005*; *Pottier et al., 2022*), potentially threatening species persistence if populations cannot evolve quickly enough or adjust via phenotypic plasticity in

structures or behaviors. This is particularly critical in highland regions, where gas conductance through the eggshell shows limited variation. While much focus has been placed on habitat structure and interspecific interactions in studies of elevational range limits, our research suggests that environmental factors affecting avian eggs, hence avian fitness, deserve further consideration.

## Methods

We searched for nests at three biological stations along an elevational gradient in the Andes of Manu National Park, Cusco Department, Peru (*Londoño et al., 2015*): Pantiacolla (380–500 m; 12°39′ S, 71°13′ W), San Pedro (1300–1600 m; 13°03′ S, 71°32′ W), and Wayquecha (2550–3200 m; 13°10′ S, 71°35′ W). The mean ± SD ambient temperatures at these sites were: lowlands = 22.5 ± 2.9°C, mid-elevation=17.6 ± 2.9°C, and highlands = 10.7 ± 3.6°C. We confirm the general expectation that relative humidity (RH) decreases with elevation (*Körner, 2007*; see Introduction) based on unpublished HOBO Pro V2 temperature/RH data (GAL). During the breeding season, nesting birds at high elevations experienced comparatively drier conditions, with 15% of measurements below 80% RH, while mid-elevation and lowland sites consistently exhibited values above 90% RH. In addition, we sampled three sites in Colombia: Remedios in the mid-Magdalena Valley, Antioquia (500 m; 6°07′ N, 74°37′ W); Estación Biológica ICESI in Parque Nacional Natural Farallones de Cali (2200–2500 m; 3°26′ N, 76°39′ W); and Cerro de Montezuma in Parque Nacional Natural Tatamá, Risaralda (1000–3000 m; 5°23′ N, 76°08′ W). All animal use protocols were approved by the University of Florida IACUC (Protocol #201106068). Field research in Peru was authorized by Conservación Amazónica ACCA and conducted under permits from the Government of Peru (Permit 0239-2013 MINAGRI-DGFFS/DGEFFS). In Colombia, protocols were approved by Universidad Icesi and authorized by ANLA (Resolution #0509, May 21, 2014).

Our sample size reflects the species for which we were able to locate nests along the elevational gradient. The uneven representation of species across elevations mirrors natural patterns of species richness and turnover, including the greater diversity of certain clades at lower elevations (e.g. antbirds), rather than a deliberate sampling bias. Furthermore, finding nests is not an easy task, and the probability of finding nests varies among species depending on nest location, nesting bird activity patterns, among others. Because a central goal of this study is to examine why particular lineages are restricted to specific elevations, we retained all available species rather than artificially balancing sample sizes through exclusions. We acknowledge that uneven sampling may influence the detection of patterns in some eggshell traits and address these potential limitations in the discussion. All comparative analyses incorporated phylogenetic corrections to account for non-independence among species.

### Gas exchange experiments

We examined variation in eggshell gas diffusion rates in 141 eggs (one egg per nest) from 108 bird species (56 species from Peru and 52 from Colombia; *Supplementary file 1*, *Figure 1*) by measuring daily water loss under known conditions of temperature, humidity, and barometric pressure (*Ar et al., 1974*). We used whole eggs, which may more accurately (than using eggshell fragments) estimate gas flux under natural conditions. This is because eggshell thickness, pore density, and pore size can vary across regions of the egg (*Rokitka and Rahn, 1987*), and the inner shell membrane also plays a role in water balance (*Paganelli et al., 1973*).

Eggs were placed in desiccators containing silica gel ($SiO_2$) to maintain near-zero humidity. Although it was not possible to strictly control temperature due to logistical constraints in remote field sites, desiccators were kept at relatively stable ambient temperatures at each station (mean ± SD: lowland = 22.9 ± 2.9°C; mid-elevation=17.7 ± 3.5°C; highland = 12.6 ± 2.4°C), which are too low to allow embryo development (*Funk and Biellier, 1944*).

Because avian eggs lose mass during incubation through evaporation (*Romanoff and Romanoff, 1949*; *Deeming, 2002*), the rate of mass loss can be used to estimate eggshell gas permeability (i.e. conductance; sensu *Ar et al., 1974*), provided ambient water vapor pressure is known. We weighed freshly laid eggs (<5 days post-laying) from nests with known clutch age that were either abandoned or partially predated, daily for 7–9 days using a MyWeight balance accurate to 0.01 g. Most eggs (approximately 80%) were 1–3 days of age; the remaining 20% lacked precise laying dates but showed

no signs of advanced embryonic development – only blood vessels were visible (*Fierro-Calderón et al., 2021*).

Water vapor conductance ($G_{H2O}$ in mg·day⁻¹ torr⁻¹; *Ar et al., 1974*) was calculated as the daily rate of water loss ($M_{H2O}$ [mg·day⁻¹]; i.e. slope of the linear regression), divided by $\Delta P_{H2O}$, corresponding to the water vapor pressure difference across the shell (torr), as diffusion rates are inversely proportional to total pressure (*Paganelli et al., 1971*). We did not detect temporal variation in water vapor conductance during our experiments, based on linear regression analyses ($R^2 > 0.8$) for all species. We discarded the data from eggs that showed unusual weight loss, usually due to eggshell fractures. Since the desiccator atmosphere is effectively at zero humidity and to ensure comparability with other studies, the water vapor pressure difference was assumed to be equal to the saturation vapor pressure of water at the egg's temperature (23.77 torr at 25°C; *Portugal et al., 2010*). Because each species was studied at only one field station (i.e. not across different elevations), we calculated mean conductance for species with more than one egg (N>1, with a maximum of three eggs per species from different nests).

## Characterizing eggshell structure

We examined the eggshell structure of bird eggs from Peru using SEM. Eggshell fragments (~3 × 3 mm²) were photographed with microscopes Philips XL30 SEM and Hitachi S3700N at the Smithsonian National Museum of Natural History (Washington, DC, USA). Structural variables (see below) were measured in 129 species (one egg per species), representing 30 families along the elevational gradient (*Supplementary file 2*). To observe the eggshell surface under natural conditions, we did not clear or stain the samples (*Fecheyr-Lippens et al., 2015*).

### Shell thickness

We acquired the shell cross-section in 129 species (*Figure 2a, c, and d*). Using ImageJ (*Rasband, 2008*), we took three independent linear measurements of shell thickness per specimen and used the mean value as our estimate of eggshell thickness. To avoid bias related to embryonic development, we did not include the cones in the mammillary layer in our measures, as their thickness may decrease during development (*Karlsson and Lilja, 2008*; *Osterström and Lilja, 2012*).

### Pores

To estimate pore density and size, we obtained the surface of three eggshell fragments from the equatorial region of each egg (*Figure 2b, d, and f*) in 116 species (one per species) in which pores were visible. We excluded several non-passerine species because surface pores were obscured by cuticle structures (*Mikhailo, 1997*; *D'Alba et al., 2016*). We developed an image-processing algorithm in MATLAB (2013a, The MathWorks, Inc, Natick, MA, USA) that automatically detects and quantifies pores in SEM images and measures their mean sectional area (i.e. size in µm²; Supplementary Material in the Figshare Digital Repository) in images at ×2000 magnification. We took three measurements for each image and used average values of pore numbers and size for subsequent analysis. As a proxy of eggshell porosity, we assumed that functional pores were represented by dark holes, in contrast to lighter gray holes that appeared occluded or superficial (*Rahn et al., 1979*; *Board and Scott, 1980*; *Vieco-Galvez et al., 2021*; see *Figure 2*).

To assess measurement reliability for traits measured repeatedly, we estimated repeatability (intraclass correlation coefficients) using linear mixed models implemented in rptR, with species as a random effect and three replicate measurements per species for each trait (eggshell thickness, pore number, and pore area). Confidence intervals were obtained by bootstrapping (1000 iterations).

## Data analysis

All variables were log-transformed to account for nonlinear scaling. To control for size-related effects, we used linear regression to $G_{H2O}$ against egg mass, and eggshell characteristics (thickness, pore density, pore size) against surface volume (mm³), and then used residuals from these regressions in subsequent analyses when significant. We estimated eggshell volume using an empirical approach based on egg length (L) and width (W), following *Hoyt, 1979*, with the formula $V = 0.51\ L\ W^2$, where L and W are in centimeters and V is in cubic centimeters. This method is widely applied in avian biology to estimate eggshell surface area from external egg dimensions without requiring direct measurement.

For analyses of $G_{H2O}$, we used egg mass to maintain consistency with prior studies. Because egg mass and volume are highly correlated, using one or the other should not introduce directional bias in comparative analyses (*Paganelli et al., 1974*).

We assessed whether conductance and eggshell traits exhibited phylogenetic signal (*Pagel, 1999*; *Freckleton et al., 2002*) and tested whether their patterns were consistent with selection at high elevations. This allowed us to characterize the mode of trait evolution (see below) and select the most appropriate evolutionary model for testing trait variation with elevation. To build the phylogeny, we randomly sampled 1000 phylogenetic trees from BirdTree (*Jetz et al., 2012*), using the *Hackett et al., 2008*, backbone. Then, we used TreeAnnotator (*Bouckaert et al., 2014*) to obtain a maximum clade-credibility tree. Then, we applied the fitContinuous function from the GEIGER package in R (*Harmon et al., 2008*) to estimate maximum-likelihood parameters for four models of trait evolution: Brownian motion (*Felsenstein, 1973*), Pagel's $\lambda$ (*Pagel, 1999*), early burst (EB; *Harmon et al., 2010*), and the Ornstein-Uhlenbeck (OU) process representing a constrained random walk with stabilizing selection toward an optimum (*Butler and King, 2004*). Model comparisons were based on Akaike's information criterion (AIC). Because gas conductance across eggshells may be influenced by nest environment (*Portugal et al., 2014*), we minimized the effect of nest type on our analysis by sampling species with ground, cup, dome, and cavity nests in equal proportions across elevations. Since conductance had low phylogenetic signal (see below), we used generalized linear models to assess the effects of egg weight (mass), elevation (station), nest type, and latitude (i.e. country) on conductance, and ranked models using AICc corrected for small sample sizes.

We compared the residual (i.e. size-independent) effects of eggshell traits across elevation categories using phylogenetic ANOVA, as implemented in the caper package in R (*Orme, 2013*). Since the results were consistent when data from Peru and Colombia were analyzed separately and combined, here, we present the analyses that included combined data from Peru and Colombia, classified by elevations in three categories: lowland (ca. 340–850 m), mid-elevation (ca. 1200–2000 m), and highland (ca. 2500–3000 m).

To test whether variation in eggshell traits is consistent with adaptation to elevational differences, we examined the relationship between residual (mass-independent) variables and elevation using PGLS models implemented in the phytools package in R (*Revell, 2010*; *Revell, 2012*). In addition to examining patterns across the complete dataset, we evaluated whether similar patterns existed within different taxonomic scales. We examined relationships between elevation and eggshell variables separately in families having data for >2 species at different elevations: Caprimulgidae (5 species), Emberizidae (4 species), Furnariidae (8 species), Grallariidae (3 species), Thraupidae (12 species), Trochilidae (13 species), Turdidae (genus *Turdus*, 4 species), and Tyrannidae and allies (19 species). Additionally, we tested for cross-elevation trait variation among individuals within one species of Passerellidae (black-faced brush finch, *A. melanolaemus*) and one species of Tyrannidae (Cinnamon Flycatcher, *P. cinnamomeus*), for which we found nests over a relatively broad elevational range (n=13 nests, range 2300–3000 m for *A. melanolaemus*; n=19 nests, range 1300–3000 m for *P. cinnamomeus*) (*Supplementary file 3*).

## Acknowledgements

We thank the ~300 assistants from ~30 countries who spend long hours in the field searching and monitoring bird nests in the Manu Bird Project for their invaluable help in the field in Colombia and Peru, especially Laura Gómez, Andres Chinome, Mario Loaiza, Tim Forrester, and Camilo Florez. In Peru, we greatly appreciate the assistance of Marianne van Vlaardingen and the Pantiacolla Lodge, Daniel Blanco and the Cock-of-the-Rock Lodge, and Conservación Amazónica – ACCA. We also thank SERNAP for granting permission to work in the Manu Park buffer zone (0239-2013 MINAGRI-DGFFS/DGEFFS 2013). In Colombia, Michelle Tapasco and Gustavo Campuzano were instrumental in assisting with logistics. Scott Whittaker provided technical assistance at the Smithsonian National Museum of Natural History (NMNH) SEM Lab. Carla Dove and Lorian Straker, from the NMNH Feather Lab, offered valuable guidance and feedback on the study of SEM images. Christopher Milensky and Jacob Saucier assisted with sample access in the NMNH egg collection. Simón Quintero assisted with the data collection on eggshell thickness. We thank the members of the 'Laboratorio de Biología Evolutiva de Vertebrados' for enriching discussions. We are grateful to Mary C Stoddard, Mark Chappell, Robert Ricklefs, Scott Robinson, Juan A Amat, Tony D Williams, Maria C Estrada-F, Orlando Acevedo,

and anonymous reviewers for their insightful comments on previous versions of the manuscript. Our animal use protocols were approved by the University of Florida IACUC, Conservación Amazónica ACCA (Peru), and Universidad Icesi (Colombia). Funding was provided by 'Proyecto Semilla' from the Facultad de Ciencias at Universidad de Los Andes, the Francois Vuilleumier Fund from the Neotropical Ornithological Society, The American Ornithologists' Union Research Award, and Predoctoral Student Fellowships – Smithsonian Institute. Fieldwork funding for searching for avian and conducting egg experiments was provided to GAL by the Alexander Skutch Award (Association of Field Ornithologists), Dexter Fellowships in Tropical Conservation Biology, Louis Agassiz Fuertes Award (Wilson Ornithological Society), the Alexander Wetmore Award (American Ornithologists' Union), and the National Science Foundation Grant DEB-1120682.

## Additional information

### Funding

| Funder | Grant reference number | Author |
| --- | --- | --- |
| National Science Foundation | DEB-1120682 | Gustavo A Londoño |
| Neotropical Ornithological Society | Francois Vuilleumier Fund | David Ocampo |
| American Ornithologists' Union | Research Grant | David Ocampo |
| National Museum of Natural History | Graduate Student Fellowships | David Ocampo |
| Association of Field Ornithologists | Alexander Skutch Award | Gustavo A Londoño |
| Wilson Ornithological Society | Louis Agassiz Fuertes Award | Gustavo A Londoño |
| American Ornithologists' Union | Alexander Wetmore Award | Gustavo A Londoño |

The funders had no role in study design, data collection and interpretation, or the decision to submit the work for publication.

### Author contributions

David Ocampo, Conceptualization, Resources, Data curation, Formal analysis, Funding acquisition, Investigation, Visualization, Methodology, Writing – original draft, Project administration, Writing – review and editing, Software, Validation; Carlos Daniel Cadena, Conceptualization, Supervision, Funding acquisition, Validation, Writing – review and editing; Esteban Correa-Agudelo, Software, Writing – review and editing, Methodology; Marcela Hernández Hoyos, Software, Methodology; Gustavo A Londoño, Conceptualization, Data curation, Supervision, Funding acquisition, Investigation, Project administration, Writing – review and editing

### Author ORCIDs

David Ocampo ⓘ https://orcid.org/0000-0003-1597-4038
Carlos Daniel Cadena ⓘ https://orcid.org/0000-0003-4530-2478
Esteban Correa-Agudelo ⓘ https://orcid.org/0000-0003-2910-7015
Marcela Hernández Hoyos ⓘ https://orcid.org/0000-0002-9723-5417
Gustavo A Londoño ⓘ https://orcid.org/0000-0003-1896-8653

### Ethics

Our animal use protocols were approved by the University of Florida IACUC (Protocol #201106068). Field research in Peru was authorized by Conservación Amazónica ACCA and conducted under permits from the Government of Peru (Permit 0239-2013 MINAGRI-DGFFS/DGEFFS). In Colombia, protocols were approved by Universidad Icesi and authorized by ANLA (Resolution #0509, 21 May

2014). To minimize potential impacts on nesting performance, we restricted our sampling to eggs from nests with known clutch age that were either abandoned or partially predated.

## Decision letter and Author response
Decision letter https://doi.org/10.7554/eLife.85564.sa1
Author response https://doi.org/10.7554/eLife.85564.sa2

## Additional files

### Supplementary files
MDAR checklist

Supplementary file 1. Eggshell conductance data across Andean bird species.

Supplementary file 2. Eggshell structure data across Andean bird species.

Supplementary file 3. Eggshell structure data for two species distributed along the Andean elevational gradient.

Supplementary file 4. Models of evolution for the eggshell traits across Andean bird species.

### Data availability
Data and code for this paper are available in the Figshare Digital Repository https://doi.org/10.6084/m9.figshare.30182047.

The following dataset was generated:

| Author(s) | Year | Dataset title | Dataset URL | Database and Identifier |
| --- | --- | --- | --- | --- |
| Ocampo D | 2025 | Adaptive variation in avian eggshell gas conductance and structure across elevational gradients? | https://doi.org/10.6084/m9.figshare.30182047 | figshare, 10.6084/m9.figshare.30182047 |

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
