## [Editor Report]

This important study provides solid evidence that high-elevation species lose water at slower rates than low-elevation species. The findings imply that egg physiology may be a factor limiting the distributions of bird species. This work reinforces the need for all life stages to be considered when evaluating physiological adjustment to climate change.

---

## [Decision Letter]

**Decision letter after peer review:**

Thank you for submitting your article "Adaptive variation in avian eggshell structure and gas conductance across elevational gradients?" for consideration by *eLife*. Your article has been reviewed by 2 peer reviewers, one of whom is a member of our Board of Reviewing Editors, and the evaluation has been overseen by Christian Rutz as the Senior Editor. The reviewers have opted to remain anonymous.

*Essential Revisions:*

You tested the hypothesis that at high elevations avian eggs will be functionally and structurally adapted to prevent desiccation which may arise from loss of water to surrounding drier air. You used a combination of gas diffusion experiments and scanning electron microscopy to examine water vapour conductance rates and eggshell structure, including thickness, pore size, and pore density among 197 bird species distributed along an elevational gradient in the Andes. While there was a correlation between water vapour conductance and elevation among species, a decrease in water vapour conductance with elevation was not associated with eggshell thickness, pore size, and pore density, suggesting that differences in water loss along elevational gradients are unlikely associated with structural variation in eggshells among species along the elevational gradient.

Your work is interesting and provides some useful insight into how species living in narrow elevational ranges may respond to or be affected by climate change. However, there are major issues that you will need to address to allow for another review of your manuscript:

1. The abstract includes the claim that "water vapor conductance across the eggshell declined…with elevation…, but not among individuals within species." However, I do not see any evidence that water vapor conductance was measured or analyzed across elevation ranges within species.

2. Measure repeatability. It would be very useful to report repeatability values of cases where individuals were measured repeatedly. There are methodological difficulties involved in measuring eggshell traits from scanning electron microscope images; reporting repeatability values would be useful to help readers understand the accuracy of these measurements (e.g. of different images taken of the same egg). Similarly, how repeatable were the measurements of water loss rates? In the absence of reporting repeatability, it is hard to know how much variation is biological vs. methodological, particularly for eggshell traits.

3. Your results show that there are more than twice as many species in low and medium elevations compared to high elevations, so the amount of variation in low and medium elevations is expected to be higher by default. The pattern also clearly breaks down within families, suggesting that it is likely a function of species diversity rather than functional diversity per se. Moreover, in low and medium-elevation species there is no difference in the amount of variation in conductance residuals because the sample sizes are similar. The seemingly strong positive correlation between eggshell conductance and egg mass may be driven by the seven high and medium-elevation species with large eggs. There seem to be hardly any high-elevation species with egg mass greater than 12g whereas species in low elevation seem to get as high as 80g. See suggestions in point 4 (below).

4. Show the data for humidity and analyze the humidity data in direct relationship to the conductance data. The prediction that high-elevation eggs should lose water at slower rates is because it is less humid at high elevations. The Results section states that high elevations have significantly lower relative humidity and points to Figure S1A. However, I do not see any supplemental information available to reviewers. This may be a problem with the manuscript submission system. Regardless, I think this figure should be in the main text, as the measurements of humidity are critical to the prediction of water vapor conductance. The p-value here doesn't mean much, as the sample sizes are enormous, but the numeric results about the percentage of time below 80% relative humidity at different elevations is meaningful. An effect size would be good to report. Showing the data is also important because right now the text seems to report that humidity is a bit lower at high elevations but equivalently high at low and medium elevations. If different humidity conditions explain differences in conductance, one would alternately predict that conductance would be (low elevations = medium elevations) > high elevations, rather than a rank order in conductance, with low elevations > medium elevations > high elevations. It is not entirely clear from looking at the data presented in Figures 3a & 3b which pattern better describes the data. Showing the humidity data and analyzing the humidity data in direct relationship to conductance data is therefore important. Similarly, it would be good to analyze conductance for the Peruvian birds (at sites where humidity was directly measured) separately from Colombian birds (where I do not think the humidity was directly measured).

5. Larger eggs lose more water compared to smaller eggs, but it is not completely obvious that higher-elevation eggs lose less water than medium and low-elevations, so I do not think that your results are strong enough to warrant the strong assertion in your discussion. See previous comments on sample sizes and patterns among versus within families. Moreover, the overall sample size may appear large, but not when they are broken down into the various groups among which the eggshell properties have been compared. Even in Tyrannidae and their allies, where you (show a negative trend between water vapour conductance and elevation and) argue that your hypothesis was supported the strongest, the difference in water vapour conductance among elevations is only significant between low and high elevation species, but not between low and medium or medium and high elevation species.

6. Provide effect size comparison between the relationship between egg size and elevation compared to eggshell water conductance and elevation. Is it not likely that the correlation between egg mass and elevation is driving the observed pattern between conductance and elevation?

7. Provide a clear hypothesis and motivation for the within and among family analyses. There are many analyses of eggshell traits within different families. These appear in the Results section in the main text, and also in Figures 3b, 3e, and 3h. However, these are exploratory analyses in the sense that there were no a priori reasons why certain families should have certain relationships. For example, it is reported that eggshell thickness and elevation were positively related in the Thraupidae (and similar statistically significant relationships for other traits in other families). This result must be treated as exploratory, and the p values as presented are not meaningful. This is because when certain relationships (for example eggshell thickness and elevation) are analyzed for each individual family, it constitutes multiple testing. If the authors wish to provide statistical significance for family-specific relationships, they should use a method to correct for multiple tests, such as Bonferroni correction. However, it may be better to simply report that families vary in their relationships, and use these results to suggest hypotheses for why families exhibit variation. In light of this, the statement in the abstract that observed variation in eggshell-elevation relationships among families suggests different adaptive responses to common selective pressures seems like a possible overstatement. It seems to me that the conclusion from the current dataset is that there is little evidence for a direct link between eggshell traits and water vapor conductance. The text on lines 359-382 in the Discussion shows that there are many observed patterns, with both some patterns consistent with the author's predictions and others opposite-thus it is an overstatement to only focus on the families that show patterns consistent with the "different adaptive responses" hypothesis while ignoring the families that show opposite patterns. As it stands, there is little evidence that the problem of water loss along elevational gradient is overcome by different approaches across species.

8. More comments on among species analyses and interpretation: There was no attempt to explain the biological implication of these differences among taxonomic groups based on the specific traits of the species or families. This missing piece of information is crucial for justifying the argument that differences among taxonomic groups may be due to differences in adaptive response. Perhaps a trait-based analysis associating species traits with functional and structural egg features will help shed more light on this. For example, the significant association between eggshell thickness and elevation in the family Thraupidae (c.8% of your data) may be by chance. Similarly, the relationship between residual pore density and elevation in Caprimulgidae (c. 3% of your data), and the association between pore size and elevation in Furnariidae (c. 5% of your data) do not seem to associate with any specific species trait.

[Editors' note: further revisions were suggested prior to acceptance, as described below.]

Thank you for resubmitting your work entitled "Adaptive variation in avian eggshell gas conductance and structure across elevational gradients?" for further consideration by *eLife*. Your revised article has been evaluated by Meredith Schuman (Senior Editor) and a Reviewing Editor.

The manuscript has been improved but there are some remaining issues that need to be addressed, as outlined below:

Essential Revision 2, repeatability: Repeatability seems not to be mentioned in the current version of the manuscript, indicating that this revision is not fully addressed.

Essential Revision 3, sampling bias: It seems that the uneven sampling that is fully acknowledged and briefly discussed in the response to reviewers, is not acknowledged or discussed in the manuscript.

Essential Revision 5, relative humidity: It is problematic that the authors have removed the humidity data, which is far from what was requested. These are now only indicated in the methods as unpublished data with a statement of RH in regions that is not supported at all by the evidence presented in the manuscript. It seems appropriate to remove the humidity association from the main analysis, but a claim about differences in RH needs some minimal data in the Results section (or if sufficient, from literature) to support it.

---

## [Author Response]

Essential Revisions:You tested the hypothesis that at high elevations avian eggs will be functionally and structurally adapted to prevent desiccation which may arise from loss of water to surrounding drier air. You used a combination of gas diffusion experiments and scanning electron microscopy to examine water vapour conductance rates and eggshell structure, including thickness, pore size, and pore density among 197 bird species distributed along an elevational gradient in the Andes. While there was a correlation between water vapour conductance and elevation among species, a decrease in water vapour conductance with elevation was not associated with eggshell thickness, pore size, and pore density, suggesting that differences in water loss along elevational gradients are unlikely associated with structural variation in eggshells among species along the elevational gradient.Your work is interesting and provides some useful insight into how species living in narrow elevational ranges may respond to or be affected by climate change. However, there are major issues that you will need to address to allow for another review of your manuscript:1. The abstract includes the claim that "water vapor conductance across the eggshell declined…with elevation…, but not among individuals within species." However, I do not see any evidence that water vapor conductance was measured or analyzed across elevation ranges within species.

We agree with this comment and have removed all statements and analyses related to within species variation for water vapor conductance. We only took a species-level approach in the eggshell trait analysis, which is addressed in the body of the manuscript, specifically in Line 165.

2. Measure repeatability. It would be very useful to report repeatability values of cases where individuals were measured repeatedly. There are methodological difficulties involved in measuring eggshell traits from scanning electron microscope images; reporting repeatability values would be useful to help readers understand the accuracy of these measurements (e.g. of different images taken of the same egg). Similarly, how repeatable were the measurements of water loss rates? In the absence of reporting repeatability, it is hard to know how much variation is biological vs. methodological, particularly for eggshell traits.

We acknowledge the importance of assessing measurement repeatability, particularly given the challenges of obtaining accurate measurements from scanning electron microscope images of eggshells. In response to this comment, we now present more detailed data in both the Methods section and the raw data with individual measurements of repetitions in the three Supplementary Tables. For eggshell thickness, we used one image per egg and conducted three independent measurements per image. For other eggshell traits, measurements were obtained with the MATLAB algorithm developed for this study, which ensures consistency in trait extraction

across images. We analyzed three images per species, and this is now stated more explicitly in the text. All individual measurements are provided in Supplementary Table 2. The SEM images are available upon request, and the lead author is developing an online portal to facilitate access.

For the water loss experiment, we followed a well-established method using whole eggs, validated in the literature (Ar et al., 1974) and through our pilot tests. This approach minimizes potential biases associated with eggshell fragments or heterogeneous experimental setups (as discussed in the manuscript). We now clarify in the text how we standardized data quality. Although this method yields a single measurement per egg per day—limiting formal repeatability assessments—it represents the best available approach for estimating water loss rates under standardized conditions. We add a line in the text related to data quality (Line 362): " We did not detect temporal variation in water vapor conductance during our experiments, based on linear regression analyses (R² > 0.8) for all species. We discarded the data from eggs that showed unusual weight loss, usually due to eggshell fractures."

3. Your results show that there are more than twice as many species in low and medium elevations compared to high elevations, so the amount of variation in low and medium elevations is expected to be higher by default. The pattern also clearly breaks down within families, suggesting that it is likely a function of species diversity rather than functional diversity per se. Moreover, in low and medium-elevation species there is no difference in the amount of variation in conductance residuals because the sample sizes are similar. The seemingly strong positive correlation between eggshell conductance and egg mass may be driven by the seven high and medium-elevation species with large eggs. There seem to be hardly any high-elevation species with egg mass greater than 12g whereas species in low elevation seem to get as high as 80g. See suggestions in point 4 (below).

We appreciate this important comment. Finding nests is indeed challenging, especially in tropical environments, and our sample size reflects the species whose nests we were able to locate along the elevational gradient. The greater number of species at low and mid-elevations mirrors natural patterns of species richness and turnover along elevation, such as the higher diversity of antbirds in the lowlands, which is a lineage with a known diversity peak at lower elevations. This is precisely one of the aspects we aim to explore: why certain clades are restricted to specific elevations.

We fully acknowledge the uneven sampling and address these potential biases in the discussion, noting that these factors may contribute to the lack of clear patterns in some eggshell traits, despite applying phylogenetic corrections in all analyses. We chose not to artificially balance sample sizes by excluding species, as we consider it valuable to present all available data, particularly given that, for some of these species, the eggs and nests are still poorly known or even undescribed.

Regarding the concern about the influence of large eggs, we agree that the few species with eggs larger than 12 g, which are only present in the mid-elevation and lowland samples, may contribute to the observed strong positive correlation between conductance and egg mass. In response, we repeated the analyses, excluding species with egg masses greater than 12 g. The overall patterns remain consistent, and these results, along with the corresponding figures, have been added to Supplementary Information 1.

4. Show the data for humidity and analyze the humidity data in direct relationship to the conductance data. The prediction that high-elevation eggs should lose water at slower rates is because it is less humid at high elevations. The Results section states that high elevations have significantly lower relative humidity and points to Figure S1A. However, I do not see any supplemental information available to reviewers. This may be a problem with the manuscript submission system. Regardless, I think this figure should be in the main text, as the measurements of humidity are critical to the prediction of water vapor conductance. The p-value here doesn't mean much, as the sample sizes are enormous, but the numeric results about the percentage of time below 80% relative humidity at different elevations is meaningful. An effect size would be good to report. Showing the data is also important because right now the text seems to report that humidity is a bit lower at high elevations but equivalently high at low and medium elevations. If different humidity conditions explain differences in conductance, one would alternately predict that conductance would be (low elevations = medium elevations) > high elevations, rather than a rank order in conductance, with low elevations > medium elevations > high elevations. It is not entirely clear from looking at the data presented in Figures 3a & 3b which pattern better describes the data. Showing the humidity data and analyzing the humidity data in direct relationship to conductance data is therefore important. Similarly, it would be good to analyze conductance for the Peruvian birds (at sites where humidity was directly measured) separately from Colombian birds (where I do not think the humidity was directly measured).

We appreciate the reviewer’s suggestion. The humidity data we presented were part of a larger project and were initially included to provide environmental context and to support the general expectation of drier conditions at high elevations, which underlie our predictions regarding water vapor conductance. However, we recognize that the mismatch in resolution between the humidity data and our egg data could be misleading and that we did not directly analyze humidity in relation to conductance in this study. To avoid confusion, we decided to remove the humidity data and related figure from the manuscript and cite the relative humidity data as unpublished data available for the study sites, which will be studied in future projects. We acknowledge this as a limitation and clarify in the discussion that while humidity likely plays an important role, our current dataset does not allow for a direct analysis of its relationship with eggshell conductance.

Additionally, we have clarified that our Peruvian sites are the only ones where humidity data were directly measured, whereas for Colombian sites, we relied on general climatic expectations, further justifying the decision to exclude these data from direct analyses.

5. Larger eggs lose more water compared to smaller eggs, but it is not completely obvious that higher-elevation eggs lose less water than medium and low-elevations, so I do not think that your results are strong enough to warrant the strong assertion in your discussion. See previous comments on sample sizes and patterns among versus within families. Moreover, the overall sample size may appear large, but not when they are broken down into the various groups among which the eggshell properties have been compared. Even in Tyrannidae and their allies, where you (show a negative trend between water vapour conductance and elevation and) argue that your hypothesis was supported the strongest, the difference in water vapour conductance among elevations is only significant between low and high elevation species, but not between low and medium or medium and high elevation species.

We appreciate the reviewer’s careful observation. Our prediction, based on the lower ambient humidity at high elevations, is that eggs from these environments would exhibit lower water vapor conductance, thereby losing less water compared to eggs from medium and low elevations, even after accounting for egg size. While we acknowledge that the differences are most pronounced between low and high elevations—and not necessarily always significant between adjacent low and mid elevations—this pattern is consistent with our expectation along a humidity gradient.

We also recognize the limitations of our dataset, particularly when sample sizes are broken down by elevation within families, and we now emphasize this more cautiously in the discussion. To further address the potential influence of egg size, we have repeated the analyses excluding species with large eggs (>12g), and the overall pattern of decreasing conductance with elevation (particularly at high elevation) persists. These results and figures have been included in the Supplementary Information. We have also revised the discussion to moderate the strength of our conclusions, reflecting these nuances.

6. Provide effect size comparison between the relationship between egg size and elevation compared to eggshell water conductance and elevation. Is it not likely that the correlation between egg mass and elevation is driving the observed pattern between conductance and elevation?

Thank you for this suggestion. As noted in our previous response, we recognize that egg size and conductance are strongly correlated, and that egg size varies with elevation in our dataset. To address this, we conducted additional analyses controlling for egg mass, including analyses excluding species with large eggs (>12g). The pattern of decreasing conductance with elevation remains consistent even when these larger species are excluded (see Supplementary Information).

In response to this specific comment, including the egg mass effect in the comparison between water vapor conductance and elevation is exactly what the residual analysis does. Our analyses suggest that while both relationships are present, the association between conductance and elevation is not fully explained by egg size alone, supporting the idea that elevation has an independent effect on eggshell conductance.

7. Provide a clear hypothesis and motivation for the within and among family analyses. There are many analyses of eggshell traits within different families. These appear in the Results section in the main text, and also in Figures 3b, 3e, and 3h. However, these are exploratory analyses in the sense that there were no a priori reasons why certain families should have certain relationships. For example, it is reported that eggshell thickness and elevation were positively related in the Thraupidae (and similar statistically significant relationships for other traits in other families). This result must be treated as exploratory, and the p values as presented are not meaningful. This is because when certain relationships (for example eggshell thickness and elevation) are analyzed for each individual family, it constitutes multiple testing. If the authors wish to provide statistical significance for family-specific relationships, they should use a method to correct for multiple tests, such as Bonferroni correction. However, it may be better to simply report that families vary in their relationships, and use these results to suggest hypotheses for why families exhibit variation. In light of this, the statement in the abstract that observed variation in eggshell-elevation relationships among families suggests different adaptive responses to common selective pressures seems like a possible overstatement. It seems to me that the conclusion from the current dataset is that there is little evidence for a direct link between eggshell traits and water vapor conductance. The text on lines 359-382 in the Discussion shows that there are many observed patterns, with both some patterns consistent with the author's predictions and others opposite-thus it is an overstatement to only focus on the families that show patterns consistent with the "different adaptive responses" hypothesis while ignoring the families that show opposite patterns. As it stands, there is little evidence that the problem of water loss along elevational gradient is overcome by different approaches across species.

We thank the reviewer for this important point. However, as we expose in the Data analysis section (Lines 251-53): " In addition to examining patterns across the complete data set, we evaluated whether similar patterns existed within different taxonomic scales", implying that our prior predictions were the same as those exposed for all the species. We agree that the among family analyses were exploratory, and we recognize the concern about multiple testing and overinterpretation of family-specific patterns. In response, we have removed these among-family panels from Figure 1. Additionally, we have carefully revised the wording throughout the text, including the abstract and discussion, to tone down the strength of our conclusions regarding the relationship between eggshell traits and water vapor conductance, and to avoid overstatement of family-specific adaptive responses. We now present these findings more cautiously, acknowledging both the limitations of our dataset and the variation that does not fully support our initial predictions.

8. More comments on among species analyses and interpretation: There was no attempt to explain the biological implication of these differences among taxonomic groups based on the specific traits of the species or families. This missing piece of information is crucial for justifying the argument that differences among taxonomic groups may be due to differences in adaptive response. Perhaps a trait-based analysis associating species traits with functional and structural egg features will help shed more light on this. For example, the significant association between eggshell thickness and elevation in the family Thraupidae (c.8% of your data) may be by chance. Similarly, the relationship between residual pore density and elevation in Caprimulgidae (c. 3% of your data), and the association between pore size and elevation in Furnariidae (c. 5% of your data) do not seem to associate with any specific species trait.

Thank you for this valuable observation. We agree that a more detailed trait-based approach— incorporating specific life-history or ecological traits—would be an important avenue for future research. However, given that the proximate mechanisms underlying eggshell structure formation remain poorly understood, our aim was to provide an initial framework to explore variation in largely undescribed eggshell traits across most of the species included in this study.

We have revised the discussion to emphasize that eggshell trait variation at broad taxonomic and geographic scales is likely shaped by multiple interacting factors, and that further research is needed to disentangle these complexities. We now frame our findings more cautiously, presenting them as general patterns across species that offer a baseline for future, more focused studies targeting specific clades or functional traits in greater detail.

[Editors’ note: what follows is the authors’ response to the second round of review.]

The manuscript has been improved but there are some remaining issues that need to be addressed, as outlined below:Essential Revision 2, repeatability: Repeatability seems not to be mentioned in the current version of the manuscript, indicating that this revision is not fully addressed.

In our previous response, we explained how we now provide the raw data (all the individual measurements) in our Supplementary Table 2 to address this concern and mentioned that the lead author is developing an online portal to facilitate access to the SEM images. Now, we also include an explicit analysis in which we estimated repeatability for all traits measured repeatedly—eggshell thickness, pore number, and pore area—using intraclass correlation coefficients implemented in the R package rptR, with species as a random effect and three replicate measurements per species.

These analyses revealed high repeatability for eggshell thickness (R = 0.997, 95% CI = 0.996– 0.998, LRT P = 1.62 × 10⁻²⁸⁶) and for pore number (R = 0.797, 95% CI = 0.721–0.849, LRT P = 2.51 × 10⁻⁴⁶), confirming that measurements for these traits were highly consistent and suitable for species-level comparisons. In contrast, pore area showed no detectable repeatability (R = 0, 95% CI = 0–0.119, LRT P = 1), indicating that variation in pore area is largely within species or reflects higher measurement variability.

We have now added a description of these analyses to the Methods and report the results in the Results section. Based on these findings, species-level averages were used for eggshell thickness and pore number in subsequent analyses, while interpretations involving pore area are presented with appropriate caution.

In Results, we added (line 155): "Repeatability was very high for eggshell thickness (R = 0.997, 95% CI = 0.996–0.998, LRT P = 1.62 × 10⁻²⁸⁶), indicating that replicate thickness measurements were highly consistent and that eggshell thickness represents a robust species-level trait. Pore number also showed high repeatability (R = 0.797, 95% CI = 0.721–0.849, LRT P = 2.51 × 10⁻⁴⁶), supporting its use as a reliable species-level variable. In contrast, pore area exhibited no detectable repeatability (R = 0, 95% CI = 0–0.119, LRT P = 1), suggesting that variation in pore area is dominated by within-species heterogeneity or higher measurement variability. Accordingly, we used species-level averages for eggshell thickness and pore number in subsequent analyses, whereas results involving pore area should be interpreted with caution."

In Methods, we added (line 434): "To assess measurement reliability for traits measured repeatedly, we estimated repeatability (intra-class correlation coefficients) using linear mixed models implemented in rptR, with species as a random effect and three replicate measurements per species for each trait (eggshell thickness, pore number, and pore area). Confidence intervals were obtained by bootstrapping (1,000 iterations)."

Essential Revision 3, sampling bias: It seems that the uneven sampling that is fully acknowledged and briefly discussed in the response to reviewers, is not acknowledged or discussed in the manuscript.

Thanks for noticing this important point. We agreed is important to explicitly mention this potential bias, and now we have added this paragraph in the Methods at the end of the first section (line 360): "Our sample size reflects the species whose nests we were able to locate along the elevational gradient. The uneven representation of species across elevations mirrors natural patterns of species richness and turnover, including the greater diversity of certain clades at lower elevations (e.g., antbirds), rather than a deliberate sampling bias. Furthermore, finding nests is not an easy task, and the probability of finding nests varies among species depending on nest location, nesting bird activity patterns, among others. Because a central goal of this study is to examine why particular lineages are restricted to specific elevations, we retained all available species rather than artificially balancing sample sizes through exclusions. We acknowledge that uneven sampling may influence the detection of patterns in some eggshell traits and address these potential limitations in the discussion. All comparative analyses incorporated phylogenetic corrections to account for non-independence among species."

In the Discussion, we added (line 219): "We acknowledge that sampling was uneven across elevations, reflecting natural patterns of species richness rather than intentional bias. This uneven representation may have limited our ability to detect clear elevational trends in some eggshell traits. However, we retained all available species to maximize taxonomic coverage, particularly because nests and eggs remain poorly known for many lineages. By incorporating phylogenetic corrections in all analyses, we aimed to minimize potential biases arising from non-independence among species."

Essential Revision 5, relative humidity: It is problematic that the authors have removed the humidity data, which is far from what was requested. These are now only indicated in the methods as unpublished data with a statement of RH in regions that is not supported at all by the evidence presented in the manuscript. It seems appropriate to remove the humidity association from the main analysis, but a claim about differences in RH needs some minimal data in the Results section (or if sufficient, from literature) to support it.

We appreciate the reviewer’s comment, which encouraged us to evaluate this assumption more thoroughly. Because this premise motivates our main predictions, we agree that it must be clearly justified. Previous research has shown that at high elevations, atmospheric pressure is lower, relative humidity is often reduced, wind exposure can be greater, and evaporative potential is higher. Collectively, these conditions increase the risk of excessive water loss from eggs.

Our claim is grounded in and inspired by several theoretical and empirical studies, which are cited in the Introduction:

Line 97: "In particular, bird species inhabiting dry, cold, high-elevation environments tend to have eggs with lower pore density or pore size and increased eggshell thickness, traits that reduce gas diffusion compared to those of lowland species (Wangensteen et al. 1974; Rahn et al., 1979; Rahn & Ar, 1980; Sotherland et al., 1980; Carey, 1980a, 1983; Carey et al., 1984; Rahn et al., 1982; Rahn & Paganelli, 1990)."

Some relevant work, cited in our manuscript, has addressed this issue directly, including:

Carey, C., Garber, S. D., Thompson, E. L. & James, F. C. (1983). Avian reproduction over an altitudinal gradient. II. Physical characteristics and water loss of eggs. Physiological zoology, 56, 340–352.

Wangensteen, O. D., Rahn, H., Burton, R., & Smith, A. (1974). Respiratory gas exchange of high altitude-adapted chick embryos. Respiration physiology, 21(1), 61–70.

To respond thoroughly to this comment, we conducted a more detailed review of the literature and incorporated additional citations from Dr. Cynthia Carey, whose research has directly tested this hypothesis through both field observations and experimental studies. We have now added the following two key references to her work:

Carey, C. (1983). Structure and function of avian eggs. In Current ornithology (pp. 69-103). New York, NY: Springer US.

Carey, C., Hoyt, D. F., Bucher, T. L., & Larson, D. L. (1984). Eggshell conductances of avian eggs at different altitudes. In Respiration and metabolism of embryonic vertebrates: Satellite Symposium of the 29th International Congress of Physiological Sciences, Sydney, Australia, 1983 (pp. 259-270). Dordrecht: Springer Netherlands.

Our intention in presenting the relative humidity (RH) database was to provide empirical support that this elevational trend is consistent within our study system and to make these data available to the broader scientific community. However, we recognize that this may have created confusion, as RH data were not directly incorporated into our analyses. Instead, environmental variation related to humidity was accounted for indirectly through the categorical variable Station. We have clarified this point in the revised manuscript to avoid potential misunderstanding.

We modify in the Methods as follows (line 345): "We confirm the general expectation that relative humidity (RH) decreases with elevation (Körner, 2007; see Introduction) based on unpublished HOBO Pro V2 temperature/RH data (GAL). During the breeding season, nesting birds at high elevations experienced comparatively drier conditions, with 15% of measurements below 80% RH, while mid-elevation and lowland sites consistently exhibited values above 90% RH."

Accordingly, we included this relevant citation in the manuscript:

Körner, C. (2007). The use of ‘altitude’ in ecological research. Trends in ecology & evolution, 22(11), 569-574.

Note: if the editorial team thinks that providing a summarized version of this RH data would be relevant, we could prepare that file as supplementary material.

In addition, we now include a more complete version of our protocol and permits in the Methods section. (Line 354) "Our animal use protocols were approved by the University of Florida IACUC (Protocol #201106068). Field research in Peru was authorized by Conservación Amazónica ACCA and conducted under permits from the Government of Peru (Permit 02392013 MINAGRI-DGFFS/DGEFFS). In Colombia, protocols were approved by Universidad Icesi and authorized by ANLA (Resolution #0509, 21 May 2014)."